# Microstructure and Properties of Multilayer Niobium-Aluminum Composites Fabricated by Explosive Welding

Yulia N. Malyutina [1], Alexander G. Anisimov [2], Albert I. Popelyukh [1], Vasiliy S. Lozhkin [1], Anatoly A. Bataev [1], Ivan A. Bataev [1,*], Yaroslav L. Lukyanov [2] and Vladimir V. Pai [2]

1  Faculty of Mechanical Engineering and Technologies, Novosibirsk State Technical University, K. Marx Ave. 20, 630073 Novosibirsk, Russia
2  Lavrentiev Institute of Hydrodynamics, Lavrentyev Ave. 15, 630090 Novosibirsk, Russia
*  Correspondence: ivanbataev@ngs.ru; Tel.: +7-91-3913-2956

**Abstract:** In this study, a layered composite material consisting of alternating aluminum and niobium layers and cladded on both sides with titanium plates was obtained by explosive welding. Microstructure of the composite was thoroughly studied using scanning electron microscopy (SEM) and transmission electron microscopy (TEM), as well as by energy dispersive X-ray spectroscopy (EDX) and electron backscattered diffraction (EBSD). Microhardness measurements, tensile test, and impact strength test were carried out to evaluate the mechanical properties of the composite. Formation of mixing zones observed near all interfaces was explained by local melting and subsequent rapid solidification. Mixing zones at Nb/Al interfaces consisted of metastable amorphous and ultrafine crystalline phases, as well as $NbAl_3$ and $Nb_2Al$ equilibrium phases. Niobium grains near the interface were significantly elongated, while aluminum grains were almost equiaxed. Crystalline grains inside the mixing zones did not have a distinct crystallographic texture. Microhardness of Al/Nb mixing zones was in the range 546–668 HV, which significantly exceeds the microhardness of initial materials. Tensile strength and impact strength of the composite were 535 MPa and 82 J/cm$^2$, respectively. These results confirm the high bonding strength between the layers.

**Keywords:** explosive welding; multilayer composite; interface morphology; Al-Nb intermetallics

## 1. Introduction

Layered composites are a combination of two or more materials, consisting of alternating layers of a strong metal and a softer component [1]. Such materials are widely used in aerospace, automotive, biomedical and other industries, because they possess sufficiently high specific rigidity, wear resistance and oxidation resistance, excellent specific strength, appropriate coefficient of thermal expansion, fatigue resistance, etc. [2,3].

Layered metal composites can be produced by various technologies such as casting [4], powder metallurgy [5,6], cladding [7,8], welding [9,10], etc. Disadvantages of these methods are their low productivity, low process versatility, and labor intensity of work pieces production.

Explosive welding is the most effective method of joining dissimilar materials, which occurs mainly in the solid state [11–13]. The explosive welding process consists in accelerating of the flyer plate due to the detonation of explosive and its subsequent oblique a high-velocity collision with a parent plate. High pressure and temperature at the collision point cause significant plastic deformation, which leads to the formation of ultra-high-strength bonding [14,15]. One of the most important features of explosive welding is formation of high-speed jet between the plates during collision. This jet consists of a thin surface layer of welded workpieces. Thus, a layer containing contaminations and oxide films is removed from the surfaces of welded plates [16]. Joints produced by explosive welding possess a perfect set of physical and mechanical properties, and generally do not

have major metallurgical defects such as cracks, macroscopic chemical inhomogeneities, porosity, etc. [17,18] which is typical for fusion welding. This method is often used to join such dissimilar materials as Ti/Al [19], Ni/Al [20], Al/Cu [21], Ti/steel [22], Zr/Cu [23], Ta/Cu [24], Nb/Stainless steel [25] that are difficult or impossible to join by other technologies.

Among above-mentioned examples, compositions of Me/Al type (where Me is some metal), such as Ni/Al [26,27], Ti/Al [28,29], Fe/Al [30,31] are of great interest in recent years. Such composites can be used for subsequent heat treatment to form layered intermetallic composites (LICs), which are considered as advanced materials for various high-temperature applications [32,33].

Less attention in the literature is given to Nb/Al-based composites, which, in turn, have obvious advantages over titanium-, nickel-, and iron-based LICs. Intermetallics of Nb/Al system have a higher melting point (1605–1960 °C, depending on the composition), high strength at elevated temperatures, and relatively low density (4.6–7.3 g/cm$^3$, depending on the composition) [34,35]. A set of unique properties of the Nb/Al composite makes it possible to use this material in the manufacture of turbines for power plants, spacecraft and other advanced applications [2,3]. However, significant difference in physical and mechanical properties of Nb and Al makes their welding by traditional methods difficult. Currently, several studies have been carried out on structure and properties of such layered composites as Nb/Al and Ti/Al/Nb, obtained by friction stir welding [36,37], joined by roll bonding [38,39] and diffusion welding [40]. However, there is limited information about Nb/Al layered composites formed by explosive welding. Elmer et al. [41] and Palmer et al. [42] used explosively welded niobium and aluminum thin sheets as inserts for subsequent welding of bulk sheets. Carvalho et al. used a niobium plate as one of the intermediate layers for aluminum to stainless steel welding [43]. However, there are essentially no studies where structure of the interfaces between explosively welded niobium and aluminum was characterized in detail, and mechanical properties of explosively welded Nb/Al composites were discussed.

Thus, the aim of this study was to obtain by explosive welding a layered composite material consisting of niobium and aluminum alternating layers, as well as to study the microstructure and mechanical properties of such composite.

## 2. Materials and Methods

### 2.1. Materials and Welding Procedure

The composition of initial materials is provided in Table 1. Commercially pure (cp)-niobium plates (more than 99.8 wt.% Nb) of 0.2 mm thick and cp-aluminum plates (more than 99.5 wt.% Al) of 0.25 mm thick were used for welding

**Table 1.** Chemical composition of initial materials.

| Materials | Elements, (wt.)% | | | | | | | | |
|---|---|---|---|---|---|---|---|---|---|
| | **Ti** | **Al** | **Mo** | **V** | **Mn** | **Fe** | **Si** | **Nb** | **Ta** |
| Cp-Al | - | Balance | - | - | - | 0.25 | 0.2 | - | |
| Cp-Nb | - | - | - | - | - | - | - | 99.9 | 0.1 |
| VT14 | Balance | 4.5 | 3.4 | 1.2 | - | - | - | - | |

The composite consisting of three layers of niobium alternating with two layers of aluminum was fabricated according to a scheme shown in Figure 1. Due to the small thickness of Nb and Al plates, a driving plate of VT14 titanium alloy of 0.6 mm thick was used for their uniform acceleration. At the bottom, the composite was welded to another VT14 alloy plate, which was necessary for subsequent technological operations that are not described in this article. Thus, the seven-layer composite of 50 × 70 × 2 mm$^3$ in size, in which aluminum and niobium alternating layers were clad with VT14 titanium alloy plates, was obtained by a single shot welding.

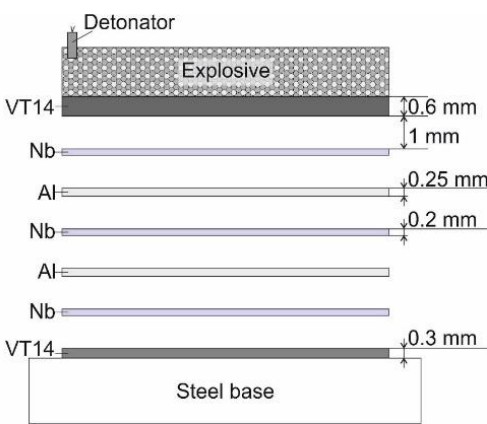

**Figure 1.** Scheme of explosive welding of 7-layered composite.

Ammonite 6ZhV with a density of 0.9 g/cm$^3$ and a detonation velocity of 2700 m/s was used as an explosive in this work. The 15 mm thick layer of explosive was applied to the surface of the driving plate. Initiation of the explosive was carried out using an electric detonator.

### 2.2. Materials Characterization Procedures

Specimens for microstructural studies were cut from the central part of the composite in such a way as to obtain a section oriented parallel to the vector of contact point velocity and normal to the surfaces of the plates. Samples were mounted into the epoxy resin, ground by abrasive papers with grit up to 2500, followed by polishing with diamond suspensions of 9, 6, 3, and 1 μm. The final polishing of the surface was carried out using suspension of colloidal silica with particle size of 0.05 μm.

Microstructural studies were performed with a Carl Zeiss EVO 50 XVP (Carl Zeiss Industrielle Messtechnik GmbH, Oberkochen, Germany) scanning electron microscope (SEM) using the backscattered electron signal, as well as with a Carl Zeiss Merlin (Carl Zeiss Industrielle Messtechnik GmbH, Oberkochen, Germany) SEM using the secondary electron signal. Titanium alloys was etched by Kroll's reagent. Elemental composition of the local zones was carried out using point and linear energy-dispersive X-ray spectroscopy (EDX) using an Oxford Instruments X-act spectrometer (Oxford Instruments, Oxon, United Kingdom) coupled with the Carl Zeiss Merlin SEM. Electron backscattered diffraction (EBSD) method was used to obtain information about phase composition of mixing zones, as well as to study crystallographic orientation of aluminum and niobium near the interface. Studies were carried out using a Carl Zeiss Sigma 300 SEM equipped with an Oxford Instruments HKL Channel 5 detector (Oxford Instruments, Oxon, United Kingdom).

Characterization of the microstructure of mixing zones was performed using a Tecnai G2 (Thermo Fisher Scientific, Waltham, Massachusetts, USA) transmission electron microscope (TEM). The specimens for TEM were disks of 3 mm in diameter cut from explosively welded samples in such a way that they contained Al/Nb interface. The TEM specimen preparation sequence was as follows: (1) mechanical grinding of samples using a SiC abrasive paper to thickness of 100 μm; (2) grinding the dimple using an Al$_2$O$_3$ suspension on Gatan 656 (Gatan, Inc., Pleasanton, California, USA) dimple grinder to thickness of 10 μm; (3) ion etching using a Gatan 691 (Gatan, Inc., Pleasanton, California, USA) precision ion polishing system with maximum beam energy of 5 keV until a through hole was formed.

### 2.3. Mechanical Properties

Vickers microhardness of composites fabricated by explosive welding was measured by a Wolpert Group 402 MVD ( YUMP, Diepoldsau, Switzerland) tester on polished transverse specimens in the direction from the driving plate to the parent plate through all layers. Load on the diamond indenter was 10 g, and dwell time was 10 sec. Spacing between each

indentation was 70 μm. For a more accurate measurement of the microhardness near the interfaces, the distance between indentation was reduced to 20 μm. The average value of microhardness was calculated from 5 measurements.

Tensile strength was determined using an Instron 3369 (Instron—division of Illinois Tool Works Inc., Glenview, Illinois, USA) universal testing machine with a maximum force of 5 kN. Samples of $3 \times 2 \times 30$ mm$^3$ in size were loaded in the direction parallel to the interfaces. Impact strength was determined using a Metrocom (Metro Com engineering s.p.a, Garbagna Novarese (No), Ytaly) impact pendulum with a maximum impact energy of 300 J. Load was applied in the direction perpendicular to the layers in samples with V-shaped notch.

The fracture surface of the specimens after mechanical testing was examined using SEM.

## 3. Results and Discussion

### 3.1. Microstructure of the Interface Region

A cross-section of the explosively welded composite consisting of titanium alloy (light gray layers), niobium (white layers) and aluminum (dark gray layers) is shown in Figure 2 made by SEM in back-scattered electron mode. Significant defects such as macrocracks, large voids or other weld discontinuities were not observed. Detailed characterization of individual layers of the composite is provided below. Although Al/Nb interfaces were of greatest interest for this study, Ti/Nb welds have also been studied.

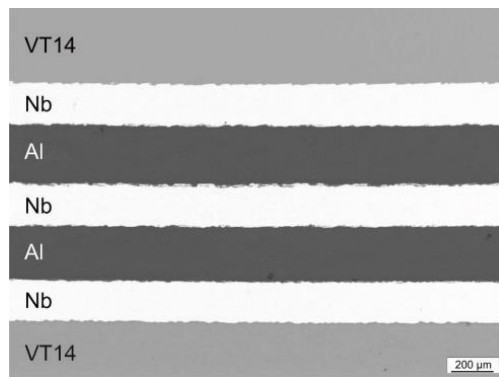

**Figure 2.** Microstructure of 7-layered composite.

### 3.2. VT14/Nb and Nb/VT4 Interfaces

The interfaces between Nb and Ti alloy in the upper and lower parts of the composite are shown in Figure 3. In both cases, one can observe formation of waves, which is a typical phenomenon for explosive welding. At the top of the composite, the waves had a less sinusoidal shape (Figure 3b) compared to the bottom Nb/VT14 interface (Figure 3d). At the same time, a more significant twisting of the material occurred in the upper part of the composite, that can be explained by dissipation of a larger amount of kinetic energy during the collision of the upper plates. The upper VT14/Nb interface also contained larger areas of localized melting. Such microvolumes are often referred to as "vortexes" due to the turbulent motion of materials during their formation (Figure 3c). At the lower interface between Nb and VT14 titanium alloy, the vortex zones were much less distinguishable.

Two types of mixing zones were formed at the interfaces between titanium alloy and niobium during explosive welding: (i) the continuous layer of 2 μm thick (Figure 3a), and (ii) localized areas separated from other mixing zones (Figure 3c,d).

On Figure 3e,f one can observe regions of severe plastic deformation resulting from high shear stresses appeared during high velocity impact. The deformed structure was observed at a distance of no more than 5 μm from the interface, indicating that shear strain during explosive welding is localized in a very narrow layer.

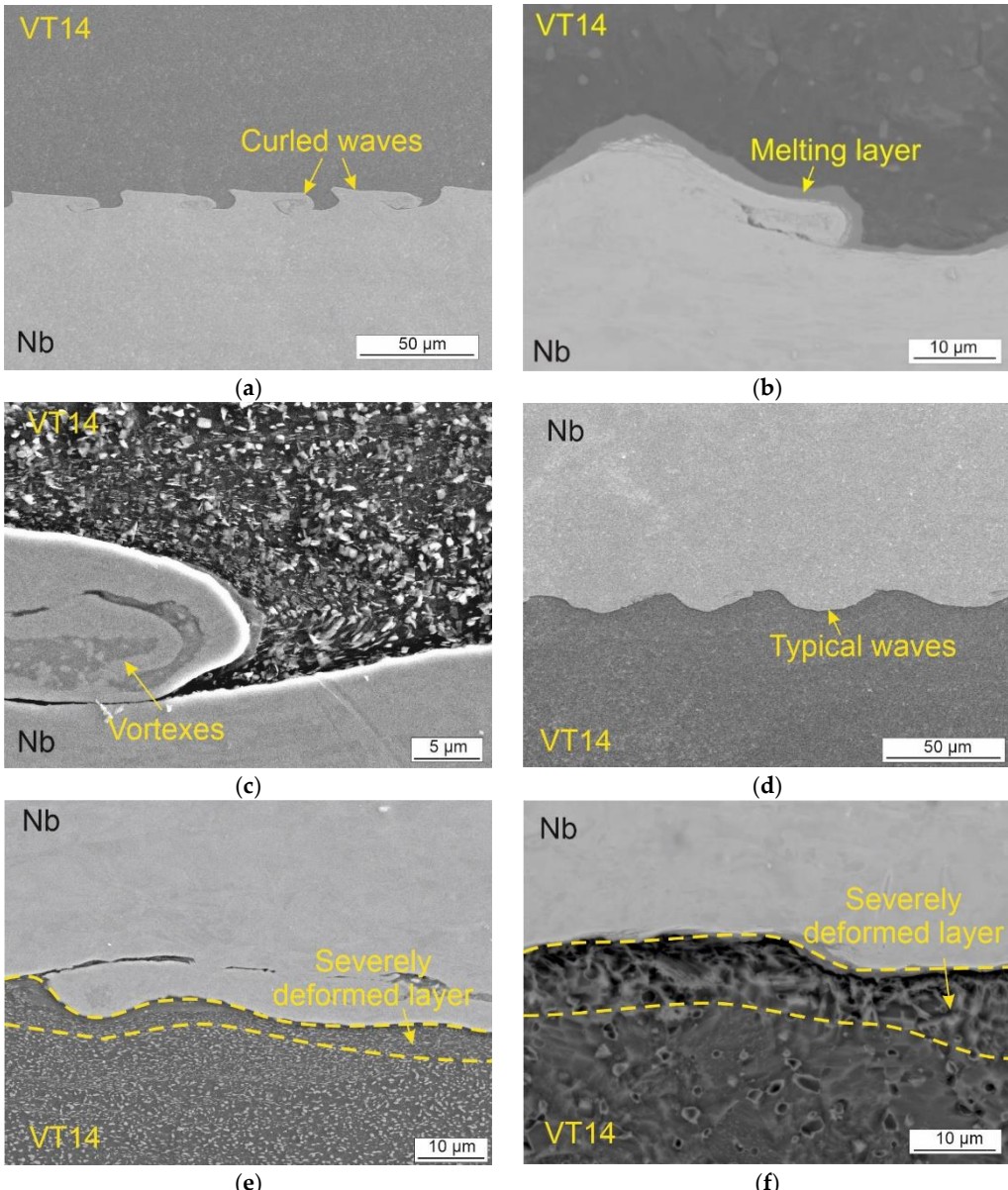

**Figure 3.** Structure of VT14/Nb (**a–c**) and Nb/VT14 (**d,f**) interfaces: (**a,b,d,e**) as-polished samples; (**c,f**) chemically etched samples.

### 3.3. Nb/Al Interfaces

Both wavy and flat interfaces were observed between niobium and aluminum. Formation of mixing zones was also typical for both of these cases (Figure 4a). In the mixing zones, local melting of welded materials was probably occurred. Despite intensive mixing, the homogenization process did not have time to complete due to the high cooling rate, and both niobium enriched and depleted regions could be observed in the mixing zones. Significant part of mixing zones consisted of intermetallic particles ranging in the size from 60 nm to 2 μm and located in a matrix of almost pure aluminum (Figure 4b). In some cases, undissolved niobium particles were observed in the mixing zones (Figure 4b).

Single cracks were found inside some areas of the solidified melt (Figure 4c). Its formation was associated simultaneously with several factors, such as high cooling rates typical for explosive welding (in some cases, the cooling rate can reach $10^5–10^7$ K/s [23]), shrinkage of melt during crystallization, mismatch of thermal expansion of welded materials, presence of intermetallic phases. The cracks propagated perpendicular to the interface

between the solidified metal and pure metals and were located within the mixing zones. Similar phenomenon was also observed in other explosively welded materials [22,44].

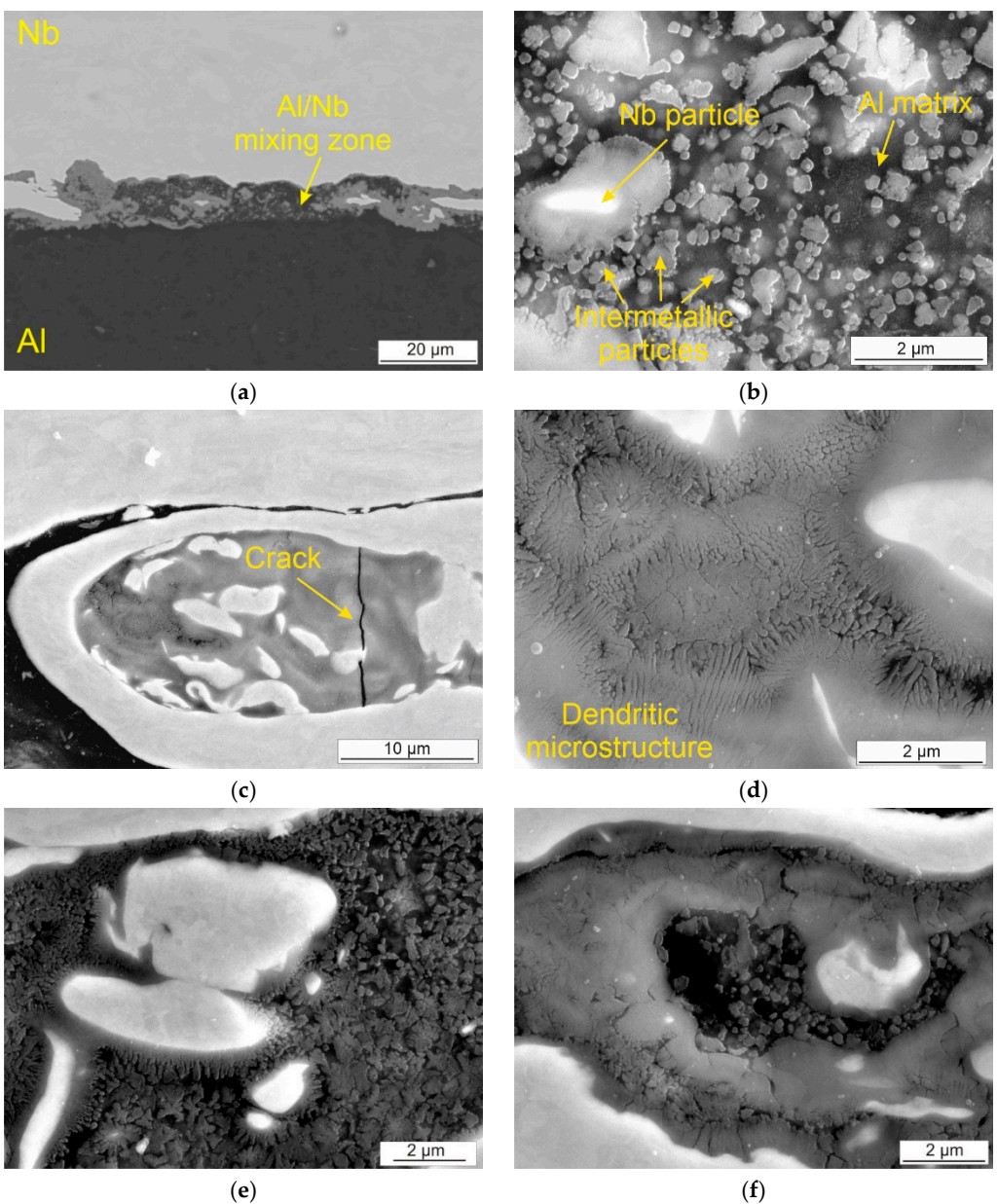

**Figure 4.** SEM images of Nb/Al interfaces formed during explosive welding: (**a**,**b**) Al/Nb mixing zone; (**c**) single crack in the solidified melt; (**d**–**f**) dendritic structure in the mixing zone.

At higher magnifications, a dendritic structure (Figure 4d–f) was observed in mixing zones, which confirms the assumption of local melting of materials during explosive welding. Similar structures were observed by Bataev et al. [20] and Kwiecen et al. [45] during explosive welding of nickel and aluminum. It's believed that one of the reasons of formation of dendritic crystals are follows. High-velocity collision of plates often results in the formation of a thermodynamically unstable melt in thin surface layers near the interface. The temperature of the liquid is much higher than that of initial metals. While the system is cooling, the liquid/solid system tends to equilibrium state by reducing its free energy. When the liquid/solid interfaces move into supercooled liquid at the temperature lower than that of the interface dendritic growth takes place on high-speed solidification. (Figure 4e). As the solid dendrite grows, latent heat of fusion is removed into the supercooled liquid, raising the temperature of the liquid to crystallization temperature. Subsequent growth

of dendrites continues until supercooled liquid is heated to crystallization temperature. Decrease in the rate of crystallization and degree of supercooling leads to transition from dendritic to planar growth with formation of crystal structure (Figure 4f).

EBSD study of Al/Nb interface is shown in Figure 5. Inverse pole figure (IPF) maps show crystals inside vortex zones that do not have predominant crystallographic orientation. Most likely, formation of randomly oriented grains was associated with intense mixing and high cooling rate in the melting zone. These conditions prevented formation of texture. According to EBSD analysis the average size of crystals formed inside mixing zone was 1.5 μm, however, this result is overestimated, since EBSD can't reliably distinguish very small grains.

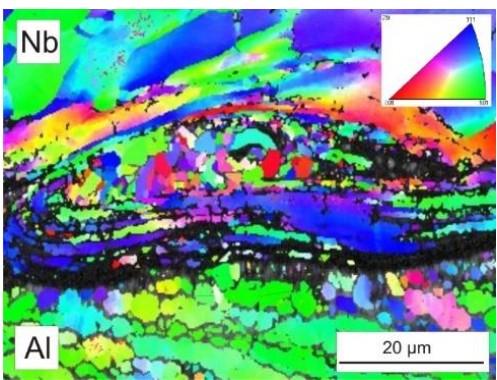

**Figure 5.** IPF-images of the mixing zone illustrating twisting of the material.

Significant structural changes caused by collision were observed close to the interfaces. Here the aluminum and niobium plates had distinct traces of severe plastic deformation. The plastic flow was especially clearly seen for niobium plates. In accordance with the applied stresses, the niobium grains were stretched along the interfaces. These grains were strongly bent in the vortex regions, which illustrates the rotational nature of the material motion during the formation of wavy boundaries.

Aluminum microstructure near the interface was represented by fine subgrains with an average size of 3 μm, slightly elongated in the direction of detonation. The formation of fine, however, rather equiaxed grains near the interface can be explained by the simultaneous action of severe plastic deformation and corresponding heating, followed by rapid transfer of the released heat to the inner part of the plates [46,47]. In contrast to niobium, the recrystallization temperature of pure aluminum is much lower, which causes recovery of the deformed material.

Results of EDX analysis of a typical mixing zone formed at an Al/Nb interface are shown in Figure 6 and in Table 2. EDX analysis was carried out both along a given line (Figure 6a), and in the areas marked in Figure 6b. The linear EDX scan indicated drastic change in chemical composition between mixing zone and initial materials (Figure 7). This is evidence that diffusion interaction between aluminum and niobium does not have time to complete due to the short duration of the welding process. Similar observations have also been made for many pairs of explosively welded dissimilar materials [48–51]. The average quantitative composition of several distinct areas is shown in Table 2. It indicates, that aluminum was not uniformly distributed in the mixing zone. Its content varied widely from 16–89 wt.%. According to Al-Nb phase diagram [52], with such aluminum content, $Nb_2Al$ and $NbAl_3$ intermetallic phases type, as well as aluminum-based solid solution can exist in the mixing zones. Here, however, it is important to note that the formation of mixing zones occurs under extremely nonequilibrium conditions, thus it is reasonable to expect formation of metastable phases. TEM was used in this study to analyze the phase constitution of the Al/Nb vortex zones, as well as to clarify data obtained using the EDX analysis.

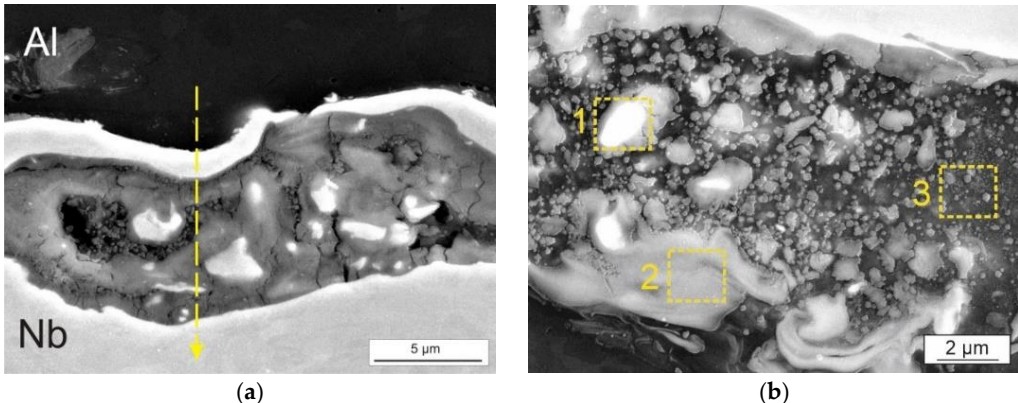

(**a**)           (**b**)

**Figure 6.** SEM images of a typical Al/Nb interface: (**a**) region used for EDX line scan, (**b**) areas used for quantitative EDX analysis.

**Table 2.** Al content in mixing zones shown in Figure 6b.

| Area | Al (wt.%) | Al (at.%) |
|---|---|---|
| 1 | 89.9 | 96.8 |
| 2 | 62.3 | 85.1 |
| 3 | 16.2 | 40.0 |

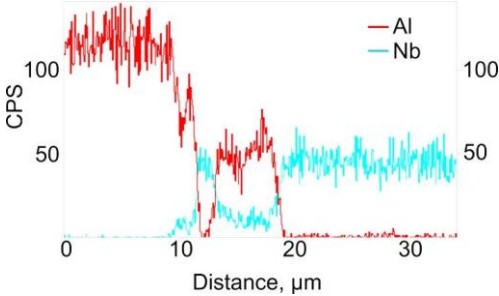

**Figure 7.** Results of EDX scan along the line shown in Figure 6a.

Figure 8a shows a TEM image of the mixing zone near its interface with aluminum. Most part of the solidified melt consisted of ultrafine-grained crystalline phases surrounded by an amorphous matrix (Figure 8b–d). Amorphous structures are frequently observed in mixing zones of explosively welded dissimilar materials [23,53]. One of the possible reasons of metallic glass formation is the rapid solidification of the melt. This is another evidence of localized melting during explosive welding.

Selected area diffraction patterns shown in Figure 8a confirm formation of NbAl$_3$ and Nb$_2$Al intermetallic phases, which could be expected by analysis of Nb-Al phase diagram. It can be seen that stoichiometric composition of the Nb$_2$Al compound (33 at.% Al, 67 at.% Nb) does not correspond to average chemical composition of vortex zone. However, since the period of liquid phase existence was short, complete homogenization in the mixing area did not occur. Therefore, due to the high cooling rates, Nb-rich phase could be retained during the crystallization of the liquid melt. An example of non-uniform distribution of niobium in aluminum in vortex zones is shown in Figure 4a.

### 3.4. Microhardness of Composite

Results of microhardness measurements are shown in Figure 9a. Average microhardness of initial materials was 260 HV for VT14, 33 HV for Al, and 118 HV for Nb.

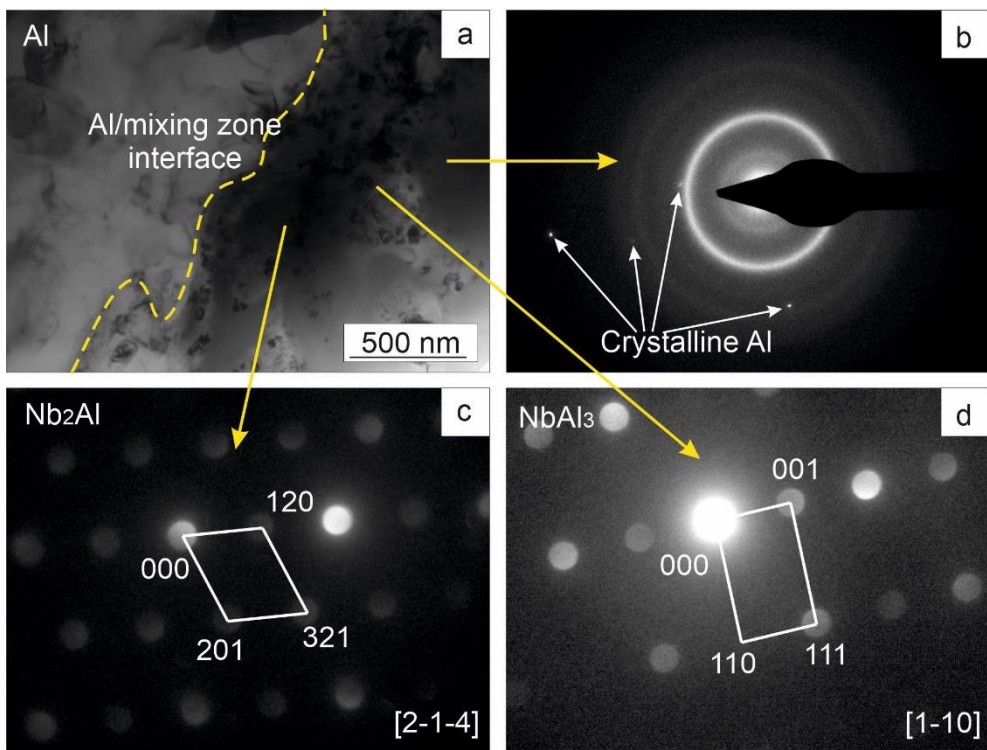

**Figure 8.** (**a**) TEM image of mixing zone at the interface of explosively welded Al and Nb; (**b**–**d**) Electron diffraction patterns, which confirm formation of metallic glass, $Nb_2Al$ and $NbAl_3$ phases, respectively.

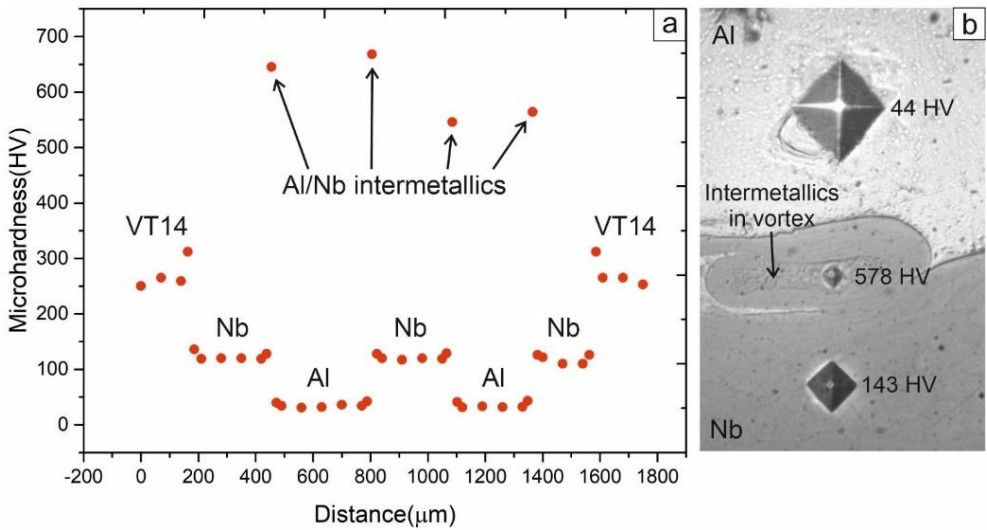

**Figure 9.** (**a**) Vickers microhardness profile across the explosively welded composite; (**b**) The image of Vickers indentations in different regions near the Al/Nb interface.

A slight increase in microhardness near the interface was observed for most of the plates. The highest microhardness value of Al near the interface was 44 HV, and that of niobium was 143 HV. For titanium alloys, the highest value of microhardness was equal to 312 HV, and it was also observed near the interface with niobium. Higher values of materials hardness near the interfaces can be explained by joint effect of strain hardening and formation of fine-grained structure in this area. According to the hardness measurements, the width of strain-hardened zone in different layers of the composite did not exceed 10–15 μm.

A sharp increase of microhardness (up to 546–668 HV) was observed inside the vortex zones formed at Al/Nb interfaces (Figure 9b), which was due to the formation of intermetallics. The microhardness values of the vortex zones are in good agreement with the data for Al-Nb-based chemical compounds reported in other works [35,54].

### 3.5. Tensile Test and Impact Test of the Composite

The interfacial bond strength of the explosively welded composite was determined by tensile strength tests. The stress-displacement curve of the composite was shown in Figure 10. Table 3 summarized the results of tensile tests of the composite, as well as the strength of the initial materials. The average value of the ultimate strength for VT14 alloy was 973 MPa. The tensile strength and yield strength of the composite exceeded the strength of pure aluminum and niobium, but was inferior to that of VT14 titanium alloy. The ultimate strength and yield strength of the composite was 535 and 336 MPa, respectively. At the same time, the composite possessed lower ductility compared to the metals in their initial state.

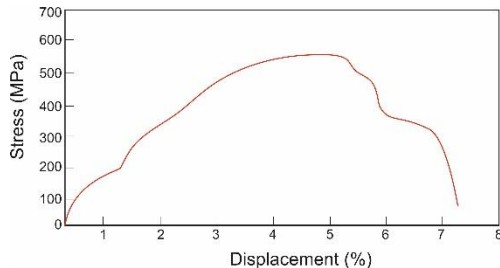

**Figure 10.** Stress-displacement curve for strength test of explosively welded composite.

**Table 3.** Mechanical properties of explosively welded composite and initial materials.

| Materials | Tensile Strength, MPa | Yield Strength, MPa | Elongation at Break, % | Impact Strength, J/sm$^2$ |
|---|---|---|---|---|
| 7-layered composite | 535 | 336 | 9 | 82 |
| Nb | 425 | 355 | 19 | 37 |
| Al | 60 | – | 24 | 110 |
| VT14 | 973 | 895 | 12 | 47 |

Fractographic analysis of the failure surface after the tensile test have shown that the fracture occurred predominantly by ductile cup and cone mechanism, as evidenced by the presence of typical fracture dimples (Figure 11a). No delamination was observed for both Nb/Ti alloy interfaces. This fact is most likely explained by the formation of ductile solid solutions between Ti and Nb, which had positive effect on the bond strength. The presence of mixing zones containing intermetallic compounds at the interfaces between aluminum and niobium led to delamination of the composite along these layers (Figure 11b). Fracture of Al/Nb intermetallic phases occurred by intergranular mechanism. Most likely, the presence of brittle intermetallic inclusions at Al/Nb interfaces adversely affected the strength of the composite as a whole.

Impact strength of the composite was 2 times higher than that of titanium alloys and niobium, yielding only to pure aluminum. An analysis of literary sources indicates that, with successful implementation of explosive welding, the resulting joints have an increased fracture toughness [55,56]. Increase of impact strength is explained by the positive effect of the interlayer boundaries. This is mainly due to the fact that delamination at the interfaces between metal and intermetallic phase prevents the main crack from propagating along the cross-section (Figure 11c). SEM studies (Figure 11d) have showed that the fracture of the composite occurred by a ductile mechanism with formation of dimples (VT14 side) and elongated cones (Al side).

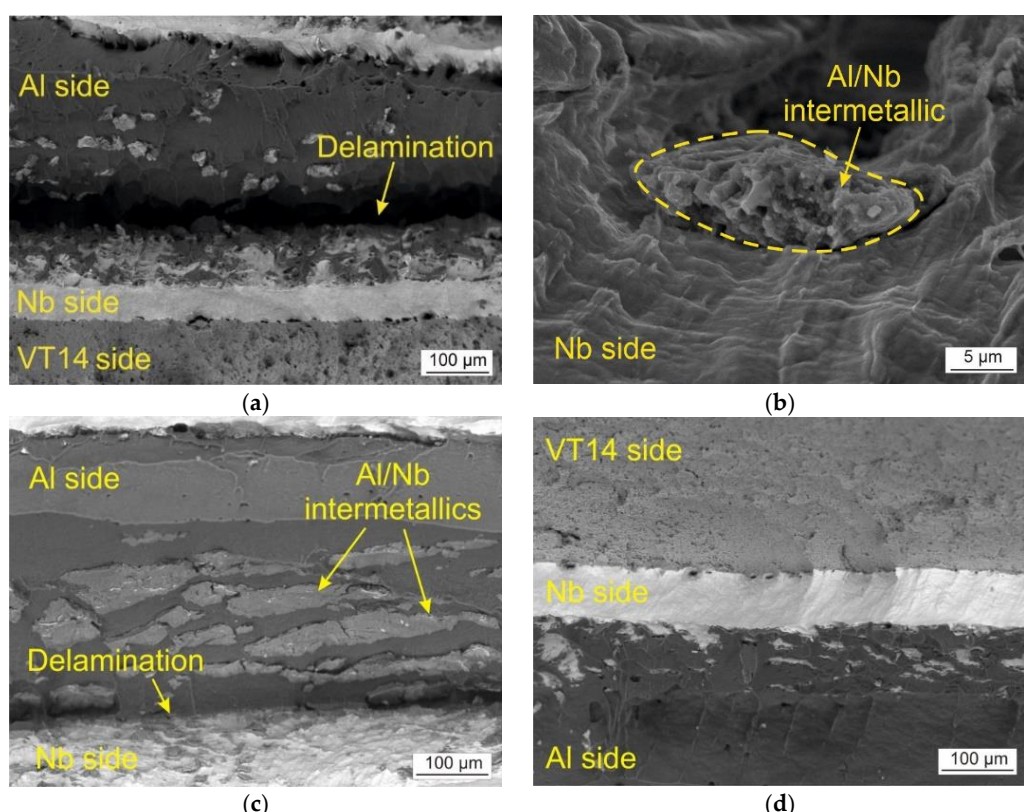

**Figure 11.** Fracture surface of explosively welded composite (**a**,**b**) after the tensile test and (**c**,**d**) after the impact test.

## 4. Conclusions

In this study, the laminated composite, consisting of titanium alloy, as well as alternating niobium and aluminum foils, was successfully fabricated by explosive welding. Microstructure and mechanical properties of the composite were studied, and the following conclusions were drawn:

1.  Materials characterization have shown that the interfaces had different appearance depending on the pair being welded. At VT14/Nb interface, a wavy morphology with formation of distinct vortex zones was observed, while Nb/VT14 had both wavy interface with discrete vortexes and nearly straight interface with formation of continuous mixing zones. Among welding defects, only rare cracks were observed in the mixing zones at Nb/Al interfaces.

2.  The mixing zones formed at Nb/Al interface possessed a nonuniform chemical composition and miscellaneous structures, which was explained by the short duration of the welding process and rapid cooling rates. TEM results confirmed formation of $NbAl_3$ and $Nb_2Al$ crystalline intermetallic phases, as well as nonequilibrium amorphous structure.

3.  EBSD analysis showed that niobium and aluminum foils near the welding boundary have a different structure: niobium grains were elongated along the interface, while aluminum consisted of small subgrains with size of 2–5 μm, which were only slightly elongated in the direction of welding.

4.  Microhardness of materials near the interface increased due to the strain hardening. The increase in microhardness to 546–668 HV inside the vortex zones formed at Al/Nb interface was explained by the formation of chemical compounds.

5.  The composite possessed satisfactory mechanical properties. In particular, the laminated structure of the composite had a positive effect on impact strength. The fracture of individual layers occurred by ductile cup and cone mechanisms. Brittle fracture

mainly occurred in the mixing zones consisted of intermetallic compounds. Formation of such compounds led to delamination at the interfaces between aluminum and niobium.

**Author Contributions:** Conceptualization, Y.N.M.; methodology, A.G.A., Y.L.L. and V.V.P.; investigation, Y.N.M., A.I.P., V.S.L. and I.A.B.; writing—original draft preparation, Y.N.M. and I.A.B.; writing—review and editing, A.A.B. and I.A.B.; supervision, Y.N.M.; project administration, Y.N.M.; funding acquisition, I.A.B. All authors have read and agreed to the published version of the manuscript.

**Funding:** This research was funded by the Federal Task of Ministry of Education and Science of the Russian Federation (project FSUN-2020-0014 (2019-0931): "Investigations of Metastable Structures Formed on Material Surfaces and Interfaces under Extreme External Impacts".

**Institutional Review Board Statement:** Not applicable.

**Informed Consent Statement:** Not applicable.

**Data Availability Statement:** The data that support the findings of this study are available from the corresponding author upon reasonable request.

**Acknowledgments:** This research was conducted at core facility "Structure, mechanical and physical properties of materials" (agreement with the Ministry of Science and Higher Education of the Russian Federation No 13.CKP.21.0034, 075-15-2021-698).

**Conflicts of Interest:** The authors declare no conflict of interest.

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
