# Peer review of "Microstructure and Properties of Multilayer Niobium-Aluminum Composites Fabricated by Explosive Welding"

_metals, doi:10.3390/met12111950_

Round 1

Reviewer 1 Report

The manuscript reports characterization of microstructure and properties of multilayer niobium-aluminum composites fabricated by explosive welding. The paper is interesting and useful, well-structured and well readable. Despite the idea of such a study is not entirely new, the manuscript does contain novel results, especially in characterization of microstructure changes e.g. near the bond zone in comparison to mechanical properties of the laminates. The presented analysis is logical, and the number of references is sufficient. In general, the presented results are of interest as for scientists and engineers. The results of the research are relatively clear but the article needs a minor revision. The only two minor comments are given:

1. Reviewer recommends adding the displacement-stress graphs for representative sample.

1. Some of the SEM microstructural observation was carried out on the etched samples. Which etching solution was used to reveal the microstructure?

Reviewer 2 Report

In this work, the authors prepared multilayer Nb/Al composites by explosive welding. Meanwhile, they characterized the microstructure and tested the mechanical properties with various methods. Both the motivation and results of the manuscript are very attractive, and the whole structure is well organized. I recommend this manuscript can be accepted if the following issues can be addressed:

1.       Can you add some applications and engineering background of Nb/Al composite in the Introduction?

2.       When performing a hardness test with an indentation space of 20 μm, is work hardening induced to affect the result? 

3.       Is Fig. 2 from SEM? Please indicate the model it used, such as SE or BSE. Moreover, please be careful when saying “Significant defects such as cracks, large voids or other weld discontinuities were not observed.” The magnification is not high enough to find defects. Only by the following figures, such as Fig. 3, the absence of defects can be evidenced. 

4.       In Fig. 3(a), the wave arrow indicates the VT 14 while it indicates the interface of Nb/VT 14. Which one is correct? The font size and capitalization in Fig. 3(e) and (f) are different from other figures. 

5.       vortexes

6.       I am not very clear about this sentence: While the system is cooling, the liquid/solid system tends to equilibrium state by reducing its free energy. Do you mean the beginning of solidification? If so, what is the relationship with the formation of dendrites? The formation of dendrites is closely related to the distribution of solute, which is not addressed in the manuscript. Please use uniform font size and capitalization in Fig. 4. 

7.       “The formation of metallic glass is … another evidence of localized melting during explosive welding”. In a strong diffusion zone, metallic glass can be found rather than only by rapid quenching. 

8.       Again in Fig. 8, please use uniform font size and capitalization.

9.       In Fig. 10(b), it should be Al/Nb Intermetallic.

10.    Please modify the reference format according to the guideline. There are too many errors in the paper and journal title. 
